METHODS AND RESOURCES

# OrthoSNAP: A tree splitting and pruning algorithm for retrieving single-copy orthologs from gene family trees

**Jacob L. Steenwyk**[1,2]*, **Dayna C. Goltz**[3], **Thomas J. Buida, III**[3], **Yuanning Li**[1,2,4],
**Xing-Xing Shen**[5], **Antonis Rokas**[1,2,6]*

**1** Vanderbilt University, Department of Biological Sciences, Nashville, Tennessee, United States of America,
**2** Vanderbilt Evolutionary Studies Initiative, Vanderbilt University, Nashville, Tennessee, United States of
America, **3** Independent Researcher, Nashville, Tennessee, United States of America, **4** Institute of Marine
Science and Technology, Shandong University, Qingdao, China, **5** Ministry of Agriculture Key Lab of
Molecular Biology of Crop Pathogens and Insects, Institute of Insect Sciences, Zhejiang University,
Hangzhou, China, **6** Heidelberg Institute for Theoretical Studies, Heidelberg, Germany

\* jacob.steenwyk@vanderbilt.edu (JLS); antonis.rokas@vanderbilt.edu (AR)

Bergen, NORWAY

**Data Availability Statement:** All results and data
presented in this study are available from figshare
(doi: 10.6084/m9.figshare.16875904).

## Abstract

Molecular evolution studies, such as phylogenomic studies and genome-wide surveys of
selection, often rely on gene families of single-copy orthologs (SC-OGs). Large gene fami-
lies with multiple homologs in 1 or more species—a phenomenon observed among several
important families of genes such as transporters and transcription factors—are often
ignored because identifying and retrieving SC-OGs nested within them is challenging. To
address this issue and increase the number of markers used in molecular evolution studies,
we developed OrthoSNAP, a software that uses a phylogenetic framework to simulta-
neously split gene families into SC-OGs and prune species-specific inparalogs. We term
SC-OGs identified by OrthoSNAP as SNAP-OGs because they are identified using a _split-
ting_ and _pruning_ procedure analogous to snapping branches on a tree. From 415,129 ortho-
logous groups of genes inferred across 7 eukaryotic phylogenomic datasets, we identified
9,821 SC-OGs; using OrthoSNAP on the remaining 405,308 orthologous groups of genes,
we identified an additional 10,704 SNAP-OGs. Comparison of SNAP-OGs and SC-OGs
revealed that their phylogenetic information content was similar, even in complex datasets
that contain a whole-genome duplication, complex patterns of duplication and loss, tran-
scriptome data where each gene typically has multiple transcripts, and contentious
branches in the tree of life. OrthoSNAP is useful for increasing the number of markers used
in molecular evolution data matrices, a critical step for robustly inferring and exploring the
tree of life.

## Introduction

Molecular evolution studies, such as species tree inference, genome-wide surveys of selection,
evolutionary rate estimation, measures of gene–gene coevolution, and others typically rely on

**Funding:** J.L.S. and A.R. were funded by the Howard Hughes Medical Institute through the James H. Gilliam Fellowships for Advanced Study program. Research in A.R.'s lab is supported by grants from the National Science Foundation (DEB-2110404), the National Institutes of Health/National Institute of Allergy and Infectious Diseases (R56 AI146096 and R01 AI153356), and the Burroughs Wellcome Fund. The funders had no role in study design, data collection and analysis, decision to publish, or preparation of the manuscript.

**Competing interests:** I have read the journal's policy and the authors of this manuscript have the following competing interests: Antonis Rokas is a scientific consultant for LifeMine Therapeutics, Inc. Jacob L. Steenwyk is a scientific consultant for Latch AI Inc.

single-copy orthologs (SC-OGs), a group of homologous genes that originated via speciation and are present in single copy among species of interest [1–6]. In contrast, paralogs—homologous genes that originated via duplication and are often members of large gene families—are typically absent from these studies (Fig 1). Gene families of orthologs and paralogs often encode functionally significant proteins such as transcription factors, transporters, and olfactory receptors [7–10]. The exclusion of SC-OGs from gene families has not only hindered our understanding of their evolution and phylogenetic informativeness but is also artificially reducing the number of gene markers available for molecular evolution studies. Furthermore, as the number of species and/or their evolutionary divergence increases in a dataset, the number of SC-OGs decreases [11,12]; case in point, no SC-OGs were identified in a dataset of 42 plants [11]. As the number of available genomes across the tree of life continues to increase, our ability to identify SC-OGs present in many taxa will become more challenging.

In light of these issues, several methods have been developed to account for paralogs in specific types of molecular evolution studies—for example, in species tree reconstruction [13]. Methods such as SpeciesRax, STAG, ASTRAL-PRO, and DISCO can be used to infer a species tree from a set of SC-OGs and gene families composed of orthologs and paralogs [11,14–16]. Other methods such as PHYLDOG [17] and guenomu [18] jointly infer the species and gene trees but require abundant computational resources, which has hindered their use for large datasets. Other software, such as PhyloTreePruner, can conduct species-specific inparalog trimming [19]. Agalma, as part of a larger automated phylogenomic workflow, can prune gene trees into maximally inclusive subtrees wherein each species, strain, or organism is represented by 1 sequence [20]. Similarly, OMA identifies subgroups of SC-OGs using graph-based clustering of sequence similarity scores [21]. Although these methods have expanded the numbers of gene markers used in species tree reconstruction, they were not designed to facilitate the retrieval of as broad a set of SC-OGs as possible for downstream molecular evolution studies such as surveys of selection. Furthermore, the phylogenetic information content of these gene families remains unknown, calling into question their usefulness.

To address this need and measure the information content of subgroups of single-copy orthologous genes, we developed OrthoSNAP, a novel algorithm that identifies SC-OGs nested within larger gene families via tree decomposition and species-specific inparalog pruning. We term SC-OGs identified by OrthoSNAP as SNAP-OGs because they were retrieved using a *s*plitti*ng a*nd *p*runing procedure. The efficacy of OrthoSNAP and the information content of SNAP-OGs was examined across 7 eukaryotic datasets, which include species with complex evolutionary histories (e.g., whole-genome duplication) or complex gene sequence data (e.g., transcriptomes, which typically have multiple transcripts per protein-coding gene). These analyses revealed OrthoSNAP can substantially increase the number of orthologs for downstream analyses such as phylogenomics and surveys of selection. Furthermore, we found that the information content of SNAP-OGs was statistically indistinguishable from that of SC-OGs suggesting the inclusion of SNAP-OGs in downstream analyses is likely to be as informative. These analyses indicate that SNAP-OGs identified by OrthoSNAP hold promise for increasing the number of markers used in molecular evolution studies, which can, in turn, be used for constructing and interpreting the tree of life.

## Results

OrthoSNAP is a novel tree traversal algorithm that conducts tree splitting and species-specific inparalog pruning to identify SC-OGs nested within larger gene families (Fig 1C). OrthoSNAP takes as input a gene family phylogeny and associated FASTA file and can output individual FASTA files populated with sequences from SNAP-OGs as well as the associated Newick tree

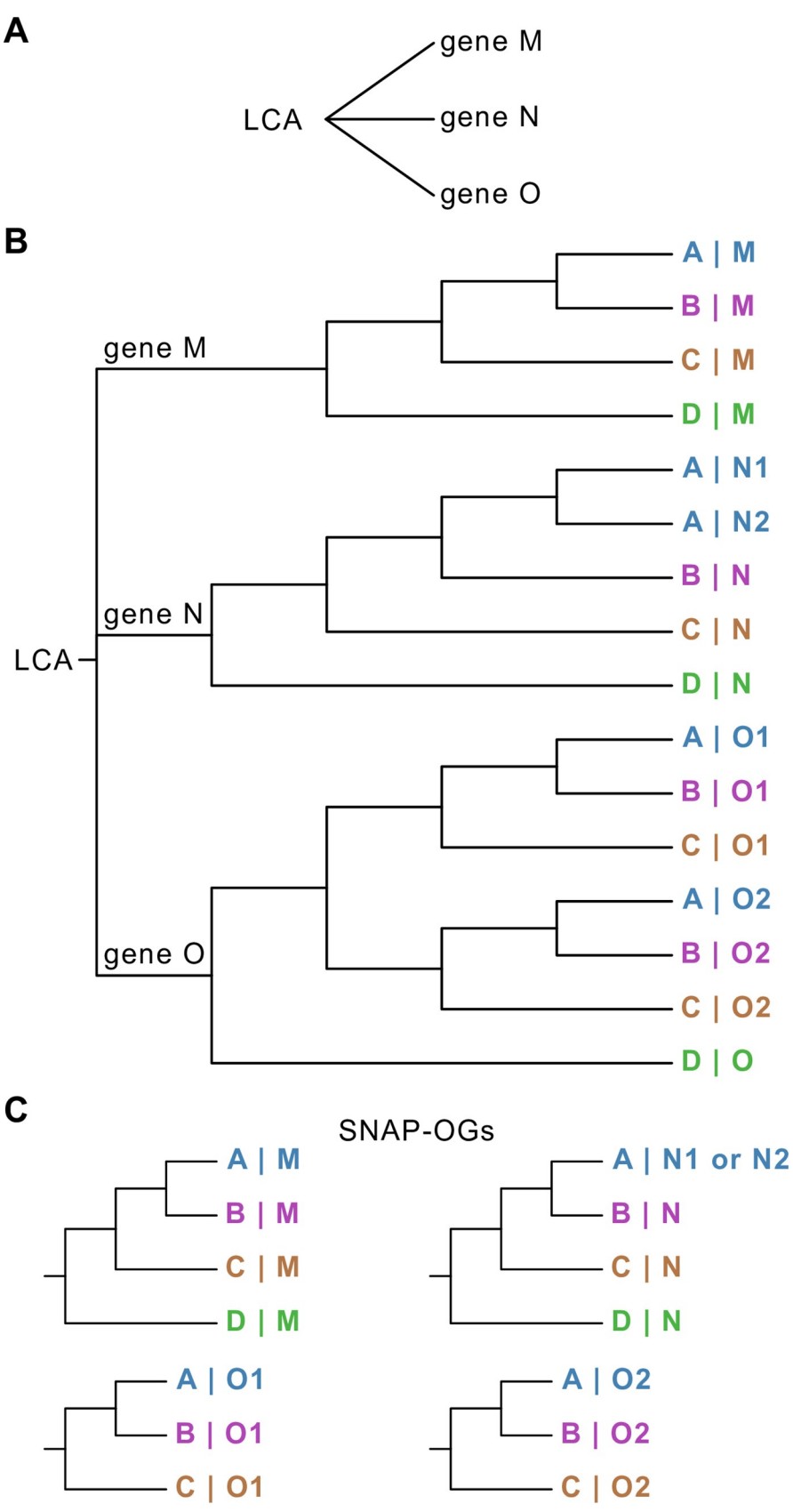

**Fig 1. Cartoon depiction of 3 classes of paralogs: outparalogs, inparalogs, and coorthologs.** (**A**) Paralogs refer to related genes that have originated via gene duplication, such as genes M, N, and O. (**B**) Outparalogs and inparalogs refer to paralogs that are related to one another via a duplication event that took place prior to or after a speciation event, respectively. With respect to the speciation event that led to the split of taxa A, B, and C from D, genes M, N, and O are outparalogs because they arose prior to the speciation event; genes O1 and O2 in taxa A, B, and C are inparalogs because they arose after the speciation event. Species-specific inparalogs are paralogous genes observed only in 1 species, strain, or organism in a dataset, such as gene N1 and N2 in species A. Species-specific inparalogs N1 and N2 in species A are also coorthologs of gene N in taxa B, C, and D; the same is true for inparalogs O1 and O2 from species A, which are coorthologs of gene O from species D. (**C**) Cartoon depiction of SNAP-OGs identified by OrthoSNAP.

files (Fig 2). During tree traversal, tree uncertainty can be accounted for by OrthoSNAP by collapsing poorly supported branches. In a set of 7 eukaryotic datasets that contained 9,821 SC-OGs, we used OrthoSNAP to identify an additional 10,704 SNAP-OGs. Using a combination of multivariate statistics and phylogenetic measures, we demonstrate that SNAP-OGs and SC-OGs have similar phylogenetic information content in all 7 datasets. This observation was consistent across datasets where the identification of large numbers of SC-OGs is challenging: flowering plants that have complex patterns of gene duplication and loss (15 SC-OGs and 653 SNAP-OGs), a lineage of budding yeasts wherein half of the species have undergone an ancient whole-genome duplication event (2,782 SC-OGs and 1,334 SNAP-OGs), and a dataset of transcriptomes where many genes are represented by multiple transcripts (390 SC-OGs and 2,087 SNAP-OGs). Lastly, similar patterns of support were observed among the 252 SC-OGs and the 1,428 SNAP-OGs in a contentious branch in the tree of life. Taken together, these results suggest that OrthoSNAP is helpful for expanding the set of gene markers available for molecular evolutionary studies, even in datasets where inference of orthology has historically been difficult due to complex evolutionary history or complex data characteristics.

## SC-OGs and SNAP-OGs have similar information content

To compare SC-OGs and SNAP-OGs, we first independently inferred orthologous groups of genes in 3 eukaryotic datasets of 24 budding yeasts (none of which have undergone whole-genome duplication), 36 filamentous fungi (*Aspergillus* and *Penicillium* species), and 26 mammals including humans, dogs, pigs, elephants, sloths, and others (S1 Table). There was variation in the number of SC-OGs and SNAP-OGs in each lineage (S1 Fig and S2 Table). Interestingly, the ratio of SNAP-OGs: SC-OGs among budding yeasts, filamentous fungi, and mammals was 0.83 (1,392: 1,668), 0.46 (2,035: 4,393), and 5.53 (1,775: 321), respectively, indicating SNAP-OGs can substantially increase the number of gene markers in certain lineages. The number of SNAP-OGs identified in a gene family with multiple homologs in 1 or more species also varied (S2 Fig).

Similar orthogroup occupancy and best-fitting models of substitutions were observed among SC-OGs and SNAP-OGs (S3 Fig and S3 Table), raising the question of whether SC-OGs and SNAP-OGs have similar information content. To answer this, the information content among multiple sequence alignments and phylogenetic trees from SC-OGs and SNAP-OGs (S4 Fig and S4 Table) was compared across 9 properties—Robinson–Foulds distance [22], relative composition variability [23], and average bootstrap support, for example—using multivariate analysis and statistics as well as information theory-based phylogenetic measures. Principal component analysis enabled qualitative comparisons between SC-OGs and SNAP-OGs in reduced dimensional space and revealed a high degree of similarity (Figs 3 and S5). Multivariate statistics—namely, multifactor analysis of variance—facilitated a quantitative comparison of SC-OGs and SNAP-OGs and revealed no difference between SC-OGs and SNAP-OGs ($p = 0.63$, F = 0.23, df = 1; S5 Table) and no interaction between the 9 properties and SC-OGs and SNAP-OGs ($p = 0.16$, F = 1.46, df = 8). Similarly, multifactor analysis of

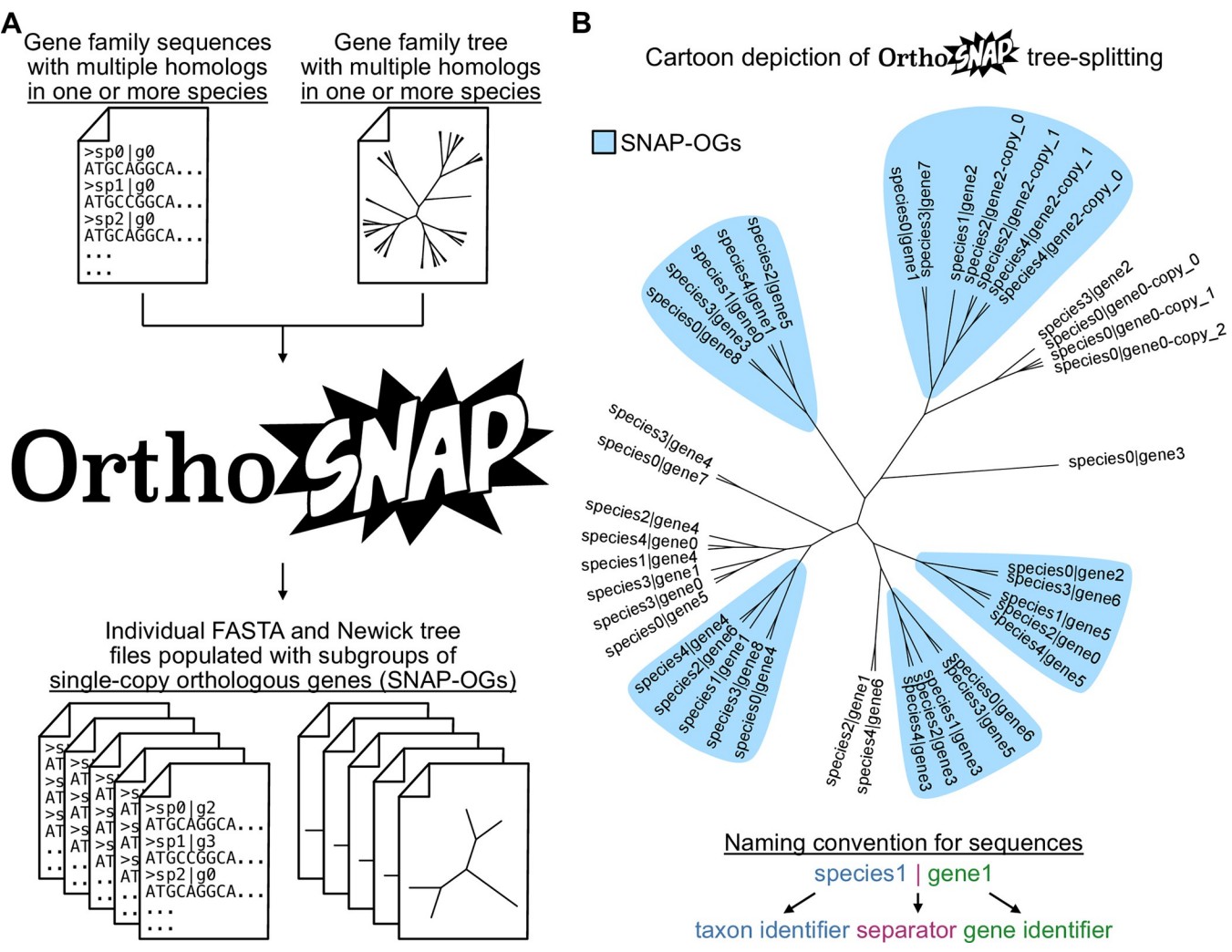

**Fig 2. Cartoon depiction of OrthoSNAP workflow.** (**A**) OrthoSNAP takes as input 2 files: a FASTA file of a gene family with multiple homologs observed in 1 or more species and the associated gene family tree. The outputted file(s) will be individual FASTA files of SNAP-OGs. Depending on user arguments, individual Newick tree files can also be outputted. (**B**) A cartoon phylogenetic tree that depicts the evolutionary history of a gene family and 5 SNAP-OGs therein. While identifying SNAP-OGs, OrthoSNAP also identifies and prunes species-specific inparalogs (e.g., species2|gene2-copy_0 and species2|gene2-copy_1), retaining only the inparalog with the longest sequence, a practice common in transcriptomics. Note, OrthoSNAP requires that sequence naming schemes must be the same in both sequences and follow the convention in which a species, strain, or organism identifier and gene identifier are separated by pipe (or vertical bar; "|") character.

variance using an additive model, which assumes each factor is independent and there are no interactions (as observed here), also revealed no differences between SC-OGs and SNAP-OGs ($p = 0.65$, F = 0.21, df = 1). Next, we calculated tree certainty, an information theory-based measure of tree congruence from a set of gene trees, and found similar levels of congruence among phylogenetic trees inferred from SC-OGs and SNAP-OGs (S6 Table). Taken together, these analyses demonstrate that SC-OGs and SNAP-OGs have similar phylogenetic information content.

We next aimed to determine if SC-OGs and SNAP-OGs have greater phylogenetic information content than a random null expectation. Groups of genes reflecting a random null expectation were constructed by randomly selecting a single sequence from representative species in multicopy orthologous genes (hereafter referred to as Random-GGs for random combinations

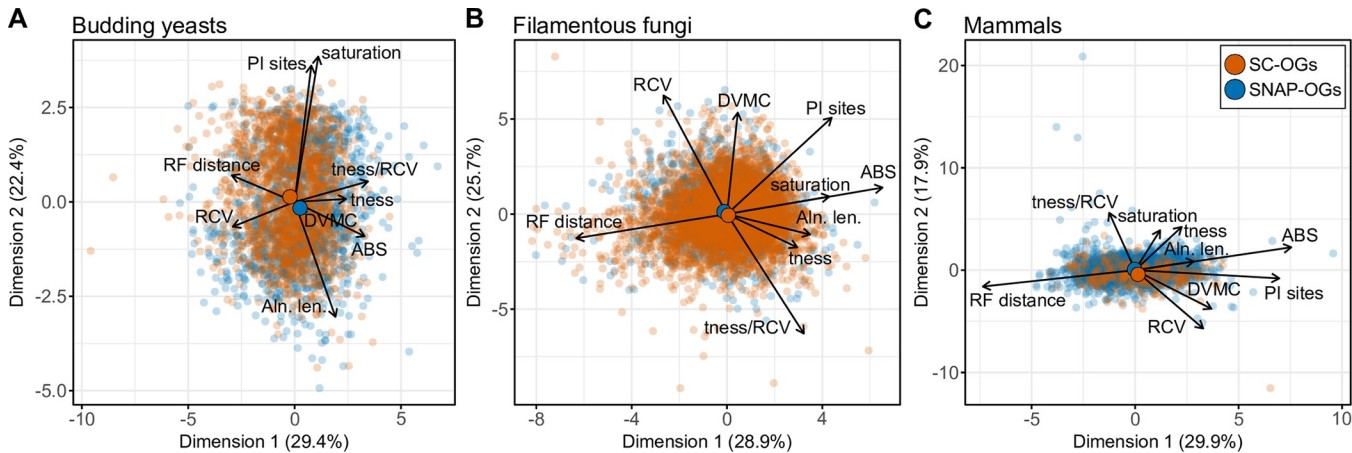

**Fig 3. SC-OGs and SNAP-OGs have similar phylogenetic information content.** To evaluate similarities and differences between SC-OGs (orange dots) and SNAP-OGs (blue dots), we examined each gene's phylogenetic information content by measuring 9 properties of multiple-sequence alignments and phylogenetic trees. We performed these analyses on 12,764 gene families from 3 datasets—24 budding yeasts (1,668 SC-OGs and 1,392 SNAP-OGs) (**A**), 36 filamentous fungi (4,393 SC-OGs and 2,035 SNAP-OGs) (**B**), and 26 mammals (321 SC-OGs and 1,775 SNAP-OGs) (**C**). Principal component analysis revealed striking similarities between SC-OGs and SNAP-OGs in all 3 datasets. For example, the centroid (i.e., the mean across all metrics and genes) for SC-OGs and SNAP-OGs, which is depicted as an opaque and larger dot, are very close to one another in reduced dimensional space. Supporting this observation, multifactor analysis of variance with interaction effects of the 6,630 SNAP-OGs and 6,634 SC-OGs revealed no difference between SC-OGs and SNAP-OGs ($p = 0.63$, F = 0.23, df = 1) and no interaction between the 9 properties and SC-OGs and SNAP-OGs ($p = 0.16$, F = 1.46, df = 8). Multifactor analysis of variance using an additive model yielded similar results wherein SC-OGs and SNAP-OGs do not differ ($p = 0.65$, F = 0.21, df = 1). There are also very few outliers of individual SC-OGs and SNAP-OGs, which are represented as translucent dots, in all 3 panels. For example, SNAP-OGs outliers at the top of panel C are driven by high treeness/RCV values, which is associated with a high signal-to-noise ratio and/or low composition bias [23]; SNAP-OG outlier at the right of panel C are driven by high average bootstrap support values, which is associated with greater tree certainty [74]; and the single SC-OG outlier observed in the bottom right of panel C is driven by a SC-OG with a high degree of violation of a molecular clock [78], which is associated with lower tree certainty [79]. Multiple-sequence alignment and phylogenetic tree properties used in principal component analysis and abbreviations thereof are as follows: average bootstrap support (ABS), degree of violation of the molecular clock (DVMC), relative composition variability, Robinson–Foulds distance (RF distance), alignment length (Aln. len.), the number of parsimony informative sites (PI sites), saturation, treeness (tness), and treeness/RCV (tness/RCV). The data underlying this figure can be found in figshare (doi: 10.6084/m9.figshare.16875904).

of orthologous and paralogous groups of genes) in the budding yeast ($N = 647$), filamentous fungi ($N = 999$), and mammalian ($N = 954$) datasets. Random-GGs were aligned, trimmed, and phylogenetic trees were inferred from the resulting multiple sequence alignments. Random-GG phylogenetic information was also calculated. Across each dataset, significant differences were observed among SC-OGs, SNAP-OGs, and Random-GGs ($p < 0.001$, F = 189.92, df = 4; Multifactor ANOVA). Further examination of differences revealed Random-GGs are significantly different compared to SC-OGs and SNAP-OGs ($p < 0.001$ for both comparisons; Tukey honest significant differences (THSD) test) in the budding yeast dataset. In contrast, SC-OGs and SNAP-OGs are not significantly different ($p = 0.42$; THSD). The same was also true for the dataset of filamentous fungi and mammals—specifically, Random-GGs were significantly different from SC-OGs and SNAP-OGs ($p < 0.001$ for each comparison in each dataset; THSD), whereas SC-OGs and SNAP-OGs were not significantly different ($p = 1.00$ for filamentous fungi dataset; $p = 0.42$ for dataset of mammals; THSD). Principal component analysis revealed Robinson–Foulds distances (a measure of tree accuracy wherein lower values represent greater tree accuracy), and relative composition variability (a measure of alignment composition bias wherein lower values represent less compositional bias), often drove differences among Random-GGs, SC-OGs, and SNAP-OGs across the datasets. In all datasets, SC-OGs and SNAP-OGs outperformed the null expectation in tree accuracy and were less compositionally biased (Table 1). These findings suggest SNAP-OGs and SC-OGs are similar in phylogenetic information content and outperform the null expectation.

**Table 1. SC-OGs and SNAP-OGs are more accurate and have less compositional biases than Random-GGs.**

| Dataset | OG type | RF distance | RCV |
|---|---|---|---|
| Budding yeasts | SC-OGs | 0.19 ± 0.12 | 0.19 ± 0.05 |
| | SNAP-OGs | 0.18 ± 0.11 | 0.18 ± 0.06 |
| | Random-GGs | **0.65 ± 0.27** | **0.27 ± 0.13** |
| Filamentous fungi | SC-OGs | 0.27 ± 0.13 | 0.12 ± 0.05 |
| | SNAP-OGs | 0.27 ± 0.12 | 0.12 ± 0.06 |
| | Random-GGs | **0.87 ± 0.11** | **0.21 ± 0.13** |
| Mammals | SC-OGs | 0.56 ± 0.22 | 0.13 ± 0.06 |
| | SNAP-OGs | 0.51 ± 0.23 | 0.11 ± 0.07 |
| | Random-GGs | **0.61 ± 0.30** | **0.15 ± 0.10** |

The first column is the dataset being examined. The second column describes the type of group of genes. The third column is the Robinson–Foulds distances, a measure of tree distance wherein higher values reflect greater inaccuracies. The fourth column is the relative composition variability, a measure of alignment composition bias wherein higher values indicate greater biases. In all datasets, SC-OGs and SNAP-OGs had better scores compared to a null expectation.

RCV, relative composition variability; RF, Robinson–Foulds distance; SC-OG, single-copy ortholog.

Values represent mean and standard deviations.

## SC-OGs and SNAP-OGs have similar performances in complex datasets

Complex biological processes and datasets pose a serious challenge for identifying markers for molecular evolution studies. To test the efficacy of OrthoSNAP in scenarios of complex evolutionary histories and datasets, we executed the same workflow described above—ortholog calling, sequence alignment, trimming, tree inference, and SNAP-OG detection—on 3 new datasets: (1) 30 plants known to have complex histories of gene duplication and loss [24–26]; (2) 30 budding yeast species wherein half of the species originated from a hybridization event that gave rise to a whole-genome duplication followed by complex patterns of loss and duplication [27–30]; and (3) 20 choanoflagellate transcriptomes, which contain thousands more transcripts than genes [31,32]; for orthology inference software, multiple transcripts per gene appear similar to artificial gene duplicates.

Corroborating previous results, OrthoSNAP successfully identified SNAP-OGs that can be used downstream for molecular evolution analyses. Specifically, using a species-occupancy threshold of 50% in the plant, budding yeast, and choanoflagellate datasets, 653, 1,334, and 2,087 SNAP-OGs were identified, respectively (Table 2). In comparison,

**Table 2. OrthoSNAP identifies SNAP-OGs in complex datasets.**

| Dataset | Challenge | Total OGs | SC-OGs (50% min. occupancy threshold) | SNAP-OGs (50% min. occupancy threshold) | SC-OGs (4 species min. threshold) | SNAP-OGs (4 species min. threshold) |
|---|---|---|---|---|---|---|
| Plants ($N$ = 30) | Evolutionary histories with extensive gene duplication and loss events | 83,034 | 15 | 653 | 200 | 15,854 |
| Budding yeasts ($N$ = 30) | Half of the species used experienced hybridization and whole-genome duplication followed by extensive loss of paralogs | 11,422 | 2,782 | 1,334 | 3,566 | 4,199 |
| Choanoflagellates ($N$ = 20) | Transcriptomes, where often multiple transcripts correspond to a single protein-coding gene | 274,028 | 390 | 2,087 | 2,438 | 11,556 |

SC-OG identification can be difficult due to complex evolutionary histories (e.g., hybridization, whole-genome duplication, and complex patterns of gene duplication and loss such as in the datasets of budding yeasts and plants) and analytical artifacts (e.g., transcriptomes with more transcripts than genes such as the choanoflagellate dataset). OrthoSNAP successfully identified SNAP-OGs in each dataset. Lowering the occupancy threshold of a SNAP-OG to a minimum of 4 enabled the identification of substantially more SNAP-OGs.

15 SC-OGs were identified in the plant dataset; 2,782 in the budding yeast dataset; and 390 in the choanoflagellate dataset. (Note that there are likely more SC-OGs than SNAP-OGs in budding yeasts because their genomes are relatively small and therefore do not have as many duplicate gene copies compared to other lineages, such as plants. Nonetheless, OrthoSNAP still substantially increases the number of markers in a phylogenomic data matrix.) To explore the impact of orthogroup occupancy, SNAP-OGs were also identified using a minimum occupancy threshold of 4 taxa. This resulted in the identification of substantially more SNAP-OGs: 15,854 in plants; 4,199 in budding yeasts; and 11,556 in choanoflagellates. Furthermore, these were substantially higher than the number of SC-OGs identified using a minimum orthogroup occupancy of 4 taxa: 200 in plants; 3,566 in budding yeasts; and 2,438 in choanoflagellates. These findings support previous observations that incorporating OrthoSNAP into ortholog identification workflows can substantially increase the number of available loci.

## SC-OGs and SNAP-OGs have similar patterns of support in a contentious branch in the tree of life

To further evaluate the information content of SNAP-OGs, we compared patterns of support among SC-OGs and SNAP-OGs in a difficult-to-resolve branch in the tree of life. Specifically, we evaluated the support between 3 hypotheses concerning deep evolutionary relationships among eutherian mammals: (1) Xenarthra (eutherian mammals from the Americas) and Afrotheria (eutherian mammals from Africa) are sister to all other Eutheria [33,34]; (2) Afrotheria are sister to all other Eutheria [35,36]; and (3) Xenarthra are sister to a clade of both Afrotheria and Eutheria (Fig 4A). Resolution of this conflict has important implications for understanding the historical biogeography of these organisms. To do so, we first obtained protein-coding gene sequences from 6 Afrotheria, 2 Xenarthra, 12 other Eutheria, and 8 outgroup taxa from NCBI (S7 Table), which represent all annotated and publicly genome assemblies at the time of this study (S8 Table). Using the protein translations of these gene sequences as input to OrthoFinder, we identified 252 SC-OGs shared across taxa; application of OrthoS-NAP identified an additional 1,428 SNAP-OGs, which represents a greater than 5-fold increase in the number of gene markers for this dataset (S8 Table). There was variation in the number of SNAP-OGs identified per orthologous group of genes (S6 Fig). The highest number of SNAP-OGs identified in an orthologous group of genes was 10, which was a gene family of olfactory receptors; olfactory receptors are known to have expanded in the evolutionary history of eutherian mammals [8]. The best-fitting substitution models were similar between SC-OGs and SNAP-OGs (S7 Fig).

Two independent tests examining support between alternative hypotheses of deep evolutionary relationships among eutherian mammals revealed similar patterns of support between SC-OGs and SNAP-OGs. More specifically, no differences were observed in gene support frequencies—the number of genes that support 1 of 3 possible hypotheses at a given branch in a phylogeny—without or with accounting for single-gene tree uncertainty by collapsing branches with low support values ($p = 0.26$ and $p = 0.05$, respectively; Fisher's exact test with Benjamini–Hochberg multitest correction; Fig 4B and S9 Table). A second test of single-gene support was conducted wherein individual gene log likelihoods were calculated for each of the 3 possible topologies. The frequency of gene-wise support for each topology was determined. No differences were observed in gene support frequency using the log likelihood approach ($p = 0.52$, respectively; Fisher's exact test). Examination of patterns of support in a contentious branch in the tree of life using 2 independent tests revealed SC-OGs and SNAP-OGs are similar and further supports the observation that they contain similar phylogenetic information.

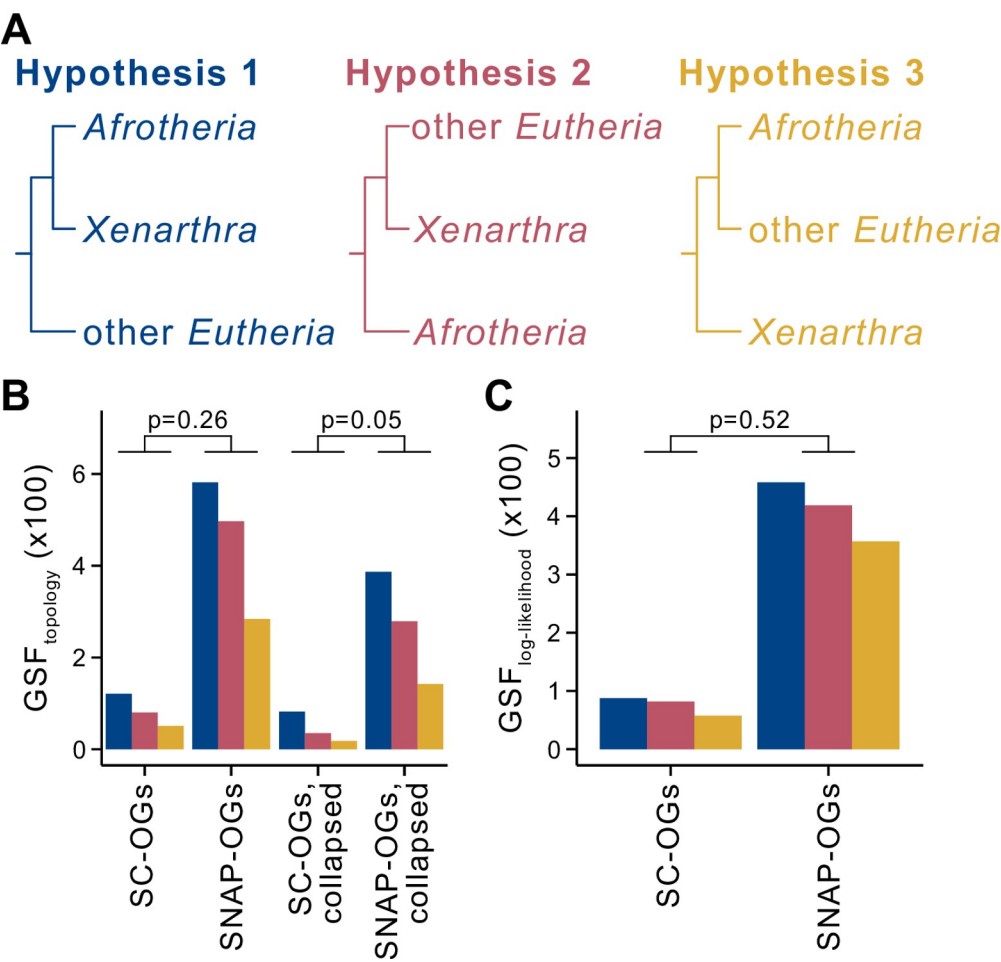

**Fig 4. SC-OGs and SNAP-OGs display similar patterns of support in a contentious branch concerning deep evolutionary relationships among eutherian mammals.** (**A**) Two leading hypotheses for the evolutionary relationships among Eutheria, which have implications for the evolution and biogeography of the clade, are that Afrotheria and Xenarthra are sister to all other Eutheria (hypothesis 1; blue) and that Afrotheria are sister to all other Eutheria (hypothesis 2; pink). The third possible, but less well-supported topology, is that Xenarthra are sister to Eutheria and Afrotheria. (**B**) Comparison of gene support frequency (GSF) values for the 3 hypotheses among 252 SC-OGs and 1,428 SNAP-OGs using an α level of 0.01 revealed no differences in support ($p$ = 0.26, Fisher's exact test with Benjamini–Hochberg multitest correction). Comparison after accounting for gene tree uncertainty by collapsing bipartitions with ultrafast bootstrap approximation support lower than 75 (SC-OGs collapsed vs. SNAP-OGs collapsed) also revealed no differences ($p$ = 0.05; Fisher's exact test with Benjamini–Hochberg multitest correction). (**C**) Examination of the distribution of frequency of topology support using gene-wise log-likelihood scores revealed no difference between SNAP-OGs and SC-OGs support for the 3 topologies ($p$ = 0.52; Fisher's exact test). The data underlying this figure can be found in figshare (doi: 10.6084/m9.figshare.16875904).

In summary, 415,129 orthologous groups of genes across 7 eukaryotic datasets contained 9,821 SC-OGs; application of OrthoSNAP identified an additional 10,704 SNAP-OGs, thereby more than doubling the number of gene markers. Comprehensive comparison of the phylogenetic information content among SC-OGs and SNAP-OGs revealed no differences in phylogenetic information content. Strikingly, this observation held true across datasets with complex evolutionary histories and when conducting hypothesis testing in a difficult-to-resolve branch in the tree of life. These findings suggest that SNAP-OGs may be useful for diverse studies of molecular evolution ranging from genome-wide surveys of selection, phylogenomic investigations, gene–gene coevolution analyses, and others.

## Discussion

Molecular evolution studies typically rely on SC-OGs. Recently, developed methods can integrate gene families of orthologs and paralogs into species tree inference but are not designed to broadly facilitate the retrieval of gene markers for molecular evolution analyses. Furthermore, the phylogenetic information content of gene families of orthologs and paralogs remains unknown. This observation underscores the need for algorithms that can identify SC-OGs nested within larger gene families, which can, in turn, be incorporated into diverse molecular evolution analyses, and a comprehensive assessment of their phylogenetic properties.

To address this need, we developed OrthoSNAP, a tree splitting and pruning algorithm that identifies SNAP-OGs, which refers to SC-OGs nested within larger gene families wherein species-specific inparalogs have also been pruned. Comprehensive examination of the phylogenetic information content of SNAP-OGs and SC-OGs from 7 empirical datasets of diverse eukaryotic species revealed that their content is similar. Inclusion of SNAP-OGs increased the size of all 7 datasets, sometimes substantially. We note that our results are qualitatively similar to those reported recently by Smith and colleagues [37], which retrieved SC-OGs nested within larger families from 26 primates and examined their performance in gene tree and species tree inference. Three noteworthy differences are that we also conduct species-specific inparalog trimming, provide a user-friendly command-line software for SNAP-OG identification, and evaluated the phylogenetic information content of SNAP-OGs and SC-OGs across 7 diverse phylogenomic datasets. We also note that our algorithm can account for diverse types of paralogy—outparalogs, inparalogs, and species-specific inparalogs—whereas other software like PhyloTreePruner, which only conducts species-specific inparalog trimming [19], and Agalma, which identifies single-copy outparalogs and inparalogs [20], can account for some, but not all, types of paralogs (S10 Table). Another difference between OrthoSNAP and other approaches is that Agalma and PhyloTreePruner both require rooted phylogenies. In contrast, OrthoSNAP will automatically midpoint root phylogenies or accept prerooted phylogenies as input. Furthermore, these algorithms are not designed to handle transcriptomic data wherein multiple transcripts per gene will be interpreted as multicopy orthologs. Thus, OrthoSNAP allows for greater user flexibility and accounts for more diverse scenarios, leading to, at least in some instances, the identification of more loci for downstream analyses (S8 Fig). Notably, these software are also different from sequence similarity graph-based inferences of subgroups of single-copy orthologous genes—such as the algorithm implemented in OMA [21]. In other words, OrthoSNAP identifies subgroups of single-copy orthologous genes by examining evolutionary histories, rather than sequence similarity values. Moreover, examination of evolutionary histories facilitates the identification of species-specific inparalogs. Finally, our results, together with other studies, demonstrate the utility of SC-OGs that are nested within larger families [15,20,37,38].

Despite the ability of OrthoSNAP to identify additional loci for molecular evolution analyses, there were instances wherein SNAP-OGs were not identified in multicopy orthologous groups of genes. We discuss 3 reasons that contribute to why SNAP-OGs could not be identified among some genes—specifically, gene families with sequence data from <50% of the taxa; gene families with complex evolutionary histories (for example, HGT and duplication/loss patterns); and gene families with evolutionary histories that differ from the species tree (for example, due to analytical factors, such as sampling and systematic error, or biological factors, such as lineage sorting or introgression/hybridization [39–41]). Notably, the first reason can, but does not always, result in inability to infer SNAP-OGs and can be, to a certain extent, addressed (e.g., by lowering the orthogroup occupancy threshold in OrthoSNAP), whereas the other 2 reasons are more challenging because they often result in a genuine absence of

SC-OGs. Furthermore, the actual number of SC-OGs (either those nested within multicopy orthologs or not) for any given group of organisms is not known, making it difficult to determine how many SNAP-OGs and SC-OGs one should expect to recover. Notably, this issue has long challenged researchers, even when ortholog identification is performed by also taking genome synteny into account [27].

Next, we discuss some practical considerations when using OrthoSNAP. In the present study, we inferred orthology information using OrthoFinder [42], but several other approaches can be used upstream of OrthoSNAP. For example, other graph-based algorithms such as OrthoMCL and OMA [21,43] or sequence similarity-based algorithms such as ortho-fisher [44] can be used to infer gene families. Similarly, sequence similarity search algorithms like BLAST+ [45], USEARCH [46], and HMMER [47] can be used to retrieve homologous sets of sequences that are used as input for OrthoSNAP. Other considerations should also be taken during the multicopy tree inference step. For example, inferring phylogenies for all orthologous groups of genes may be a computationally expensive task. Rapid tree inference software —such as FastTree or IQTREE with the "-fast" parameter [48,49]—may expedite these steps (but users should be aware that this may result in a loss of accuracy in inference; [50]).

We suggest employing "best practices" when inferring groups of putatively orthologous genes, including SNAP-OGs. Specifically, orthology information can be further scrutinized using phylogenetic methods. Orthology inference errors may occur upstream of OrthoSNAP; for example, SNAP-OGs may be susceptible to erroneous inference of orthology during upstream clustering of putatively orthologous genes. One method to identify putatively spurious orthology inference is by identifying long terminal branches [51]. Terminal branches of outlier length can be identified using the "spurious_sequence" function in PhyKIT [52]. Other tools, such as PhyloFisher, UPhO, and other orthology inference pipelines employ similar strategies to refine orthology inference [53–55]. Lastly, we acknowledge that future iterations of OrthoSNAP may benefit from incorporating additional layers of information, such as sequence similarity scores or synteny. Even though OrthoSNAP did identify SNAP-OGs in some complex datasets where synteny has previously been very helpful, such as the budding yeast dataset, other ancient and rapidly evolving lineages may benefit from synteny analysis to dissect complex relationships of orthology [51,56–58].

Taken together, we suggest that OrthoSNAP is useful for retrieving single-copy orthologous groups of genes from gene family data and that the identified SNAP-OGs have similar phylogenetic information content compared to SC-OGs. In combination with other phylogenomic toolkits, OrthoSNAP may be helpful for reconstructing the tree of life and expanding our understanding of the tempo and mode of evolution therein.

## Methods

### OrthoSNAP availability and documentation

OrthoSNAP is available under the MIT license from GitHub (https://github.com/JLSteenwyk/orthosnap), PyPi (https://pypi.org/project/orthosnap), and the Anaconda cloud (https://anaconda.org/JLSteenwyk/orthosnap). OrthoSNAP is also freely available to use via the Latch-Bio (https://latch.bio/) cloud-based console (dedicated interface link: https://console.latch.bio/explore/65606/info). Documentation describes the OrthoSNAP algorithm, parameters, and provides user tutorials (https://jlsteenwyk.com/orthosnap).

### OrthoSNAP algorithm description and usage

We next describe how OrthoSNAP identifies SNAP-OGs. OrthoSNAP requires 2 files as input: one is a FASTA file that contains 2 or more homologous sequences in 1 or more species

and the other the corresponding gene family phylogeny in Newick format. In both the FASTA and Newick files, users must follow a naming scheme—wherein species, strain, or organism identifiers and gene sequences identifiers are separated by a vertical bar (also known as a pipe character or "|")—which allows OrthoSNAP to determine which sequences were encoded in the genome of each species, strain, or organism. After initiating OrthoSNAP, the gene family phylogeny is first midpoint rooted (unless the user specifies the inputted phylogeny is already rooted) and then SNAP-OGs are identified using a tree-traversal algorithm. To do so, OrthoS-NAP will loop through the internal branches in the gene family phylogeny and evaluate the number of distinct taxa identifiers among children terminal branches. If the number of unique taxon identifiers is greater than or equal to the orthogroup occupancy threshold (default: 50% of total taxa in the inputted phylogeny; users can specify an integer threshold), then all children branches and termini are examined further; otherwise, OrthoSNAP will examine the next internal branch. Next, OrthoSNAP will collapse branches with low support (default: 80, which is motivated by using ultrafast bootstrap approximations [59] to evaluate bipartition support; users can specify an integer threshold) and conduct species-specific inparalog trimming wherein the longest sequence is maintained, a practice common in transcriptomics. However, users can specify whether the shortest sequence or the median sequence (in the case of 3 or more sequences) should be kept instead. Users can also pick which species-specific inparalog to keep based on branch lengths (the longest, shortest, or median branch length in the case of having 3 or more sequences). Species-specific inparalogs are defined as sequences encoded in the same genome that are sister to one another or belong to the same polytomy [19]. The resulting set of sequences is examined to determine if 1 species, strain, or organism is represented by 1 sequence and ensure these sequences have not yet been assigned to a SNAP-OG. If so, they are considered a SNAP-OG; if not, OrthoSNAP will examine the next internal branch. When SNAP-OGs are identified, FASTA files of SNAP-OG sequences are outputted. Users can also output the subtree of the SNAP-OG using an additional argument.

The principles of the OrthoSNAP algorithm are also described using the following pseudocode:
FOR internal branch in midpoint rooted gene family phylogeny:

> IF orthogroup occupancy among children termini is greater than or equal to orthogroup occupancy threshold;

>> Collapse poorly supported bipartitions and trim species-specific inparalogs;

>> IF each species, strain, or organism among the trimmed set of species, strains, or organisms is represented by only one sequence and no sequences being examined have been assigned to a SNAP-OG yet;

>>> Sequences represent a SNAP-OG and are outputted to a FASTA file

>> ELSE

>>> examine next internal branch

> ELSE

>> examine next internal branch

ENDFOR
To enhance the user experience, arguments or default values are printed to the standard output, a progress bar informs the user of how of the analysis has been completed, and the number of SNAP-OGs identified as well as the names of the outputted FASTA files are printed to the standard output.

## Development practices and design principles to ensure long-term software stability

Archival instabilities among software threatens the reproducibility of bioinformatics research [60]. To ensure long-term stability of OrthoSNAP, we implemented previously established rigorous development practices and design principles [44,52,61,62]. For example, OrthoSNAP features a refactored codebase, which facilitates debugging, testing, and future development. We also implemented a continuous integration pipeline to automatically build, package, and install OrthoSNAP across Python versions 3.7, 3.8, and 3.9. The continuous integration pipeline also conducts 57 unit and integration tests, which span 95.90% of the codebase and ensure faithful function of OrthoSNAP.

## Dataset generation

To generate a dataset for identifying SNAP-OGs and comparing them to SC-OGs, we first identified putative groups of orthologous genes across 4 empirical datasets. To do so, we first downloaded proteomes for each dataset, which were obtained from publicly available repositories on NCBI (S1 and S7 Tables) or figshare [51]. Each dataset varied in its sampling of sequence diversity and in the evolutionary divergence of the sampled taxa. The dataset of 24 budding yeasts spans approximately 275 million years of evolution [51]; the dataset of 36 filamentous fungi spans approximately 94 million years of evolution [63]; the dataset of 26 mammals spans approximately 160 million years of evolution [64]; and the dataset of 28 eutherian mammals—which was used to study the contentious deep evolutionary relationships among eutherian mammals—concerns an ancient divergence that occurred approximately 160 million years ago [65]. Putatively orthologous groups of genes were identified using OrthoFinder, v2.3.8 [42], with default parameters, which resulted in 46,645 orthologous groups of genes with at least 50% orthogroup occupancy (S8 Table).

To infer the evolutionary history of each orthologous group of genes, we first individually aligned and trimmed each group of sequences using MAFFT, v7.402 [66], with the "auto" parameter and ClipKIT, v1.1.3 [61], with the "smart-gap" parameter, respectively. Thereafter, we inferred the best-fitting substitution model using Bayesian information criterion and evolutionary history of each orthologous group of genes using IQ-TREE2, v2.0.6 [49]. Bipartition support was examined using 1,000 ultrafast bootstrap approximations [59].

To identify SNAP-OGs, the FASTA file and associated phylogenetic tree for each gene family with multiple homologs in 1 or more species was used as input for OrthoSNAP, v0.0.1 (this study). Across 40,011 gene families with multiple homologs in 1 or more species in all datasets, we identified 6,630 SNAP-OGs with at least 50% orthogroup occupancy (S1 Fig and S8 Table). Unaligned sequences of SNAP-OGs were then individually aligned and trimmed using the same strategy as described above. To determine gene families that were SC-OGs, we identified orthologous groups of genes with at least 50% orthogroup occupancy and each species, strain, or organism was represented by only 1 sequence—6,634 orthologous groups of genes were SC-OGs.

## Measuring and comparing information content among SC-OGs and SNAP-OGs

To compare the information content of SC-OGs and SNAP-OGs, we calculated 9 properties of multiple sequence alignments and phylogenetic trees associated with robust phylogenetic signal in the budding yeasts, filamentous fungi, and mammalian datasets (S4 Table). More specifically, we calculated information content from phylogenetic trees such as measures of tree

certainty (average bootstrap support), accuracy (Robinson–Foulds distance; [67]), signal-to-noise ratios (treeness; [68]), and violation of clock-like evolution (degree of violation of a molecular clock or DVMC; [69]). Information content was also measured among multiple sequence alignments by examining alignment length and the number of parsimony-informative sites, which are associated with robust and accurate inferences of evolutionary histories [70] as well as biases in sequence composition (RCV; [68]). Lastly, information content was also evaluated using metrics that consider characteristics of phylogenetic trees and multiple sequence alignments such as the degree of saturation, which refers to multiple substitutions in multiple sequence alignments that underestimate the distance between 2 taxa [71], and treeness/RCV, a measure of signal-to-noise ratios in phylogenetic trees and sequence composition biases [68]. For tree accuracy, phylogenetic trees were compared to species trees reported in previous studies [51,63,64]. All properties were calculated using functions in PhyKIT, v1.1.2 [52]. The function used to calculate each metric and additional information are described in S4 Table.

Principal component analysis across the 9 properties that summarize phylogenetic information content was used to qualitatively compare SC-OGs and SNAP-OGs in reduced dimensional space. Principal component analysis, visualization, and determination of property contribution to each principal component was conducted using factoextra, v1.0.7 [72], and FactoMineR, v2.4 [73], in the R, v4.0.2 (https://cran.r-project.org/), programming environment. Statistical analysis using a multifactor ANOVA was used to quantitatively compare SC-OGs and SNAP-OGs using the res.aov() function in R.

Information theory-based approaches were used to evaluate incongruence among SC-OGs and SNAP-OGs phylogenetic trees. More specifically, we calculated tree certainty and tree certainty-all [74–76], which are conceptually similar to entropy values and are derived from examining support among a set of gene trees and the 2 most supported topologies or all topologies that occur with a frequency of ≥5%, respectively. More simply, tree certainty values range from 0 to 1 in which low values are indicative of low congruence among gene trees and high values are indicative of high congruence among gene trees. Tree certainty and tree certainty-all values were calculated using RAxML, v8.2.10 [77].

To examine patterns of support in a contentious branch concerning deep evolutionary relationships among eutherian mammals, we calculated gene support frequencies and ΔGLS. Gene support frequencies were calculated using the "polytomy_test" function in PhyKIT, v1.1.2 [52]. To account for uncertainty in gene tree topology, we also examined patterns of gene support frequencies after collapsing bipartitions with ultrafast bootstrap approximation support lower than 75 using the "collapse" function in PhyKIT. To calculate gene-wise log likelihood values, partition log-likelihoods were calculated using the "wpl" parameter in IQ-TREE2 [49], which required as input a phylogeny in Newick format that represented either hypothesis 1, 2, or 3 (Fig 4A) and a concatenated alignment of SC-OGs and SNAP-OGs with partition information. Thereafter, the log likelihood values were used to assign genes to the topology they best supported. Inconclusive genes, defined as having a gene-wise log likelihood difference of less than 0.01, were removed.

The same methodologies—orthology inference, multiple-sequence alignment, trimming, tree inference, SNAP-OG identification, and phylogenetic information content calculations—were also applied to 3 additional datasets that represent complex datasets. Specifically, 30 plants (with a history of extensive gene duplication and loss events), 30 budding yeast species (15 of which experienced whole-genome duplication), and 20 choanoflagellate transcriptomes (where typically multiple transcripts correspond to a single protein-coding gene) [31,32].

## Supporting information

**S1 Fig. Numbers of orthogroups, single-copy orthogroups, orthogroups with 1 or more homologs in 1 species, and the number of SNAP-OGs identified for each dataset.** (**A**) The total number of orthogroups with at least 50% ortholog occupancy for each dataset. (**B**) The number of single-copy orthologs (SC-OGs) for each dataset (with at least 50% taxon occupancy). (**C**) The number of multicopy orthologs (or orthologous groups of genes wherein 1 or more species is represented by 2 or more sequences; MC-OGs) for each dataset (with at least 50% taxon occupancy). (**D**) The number of SNAP-OGs identified in each dataset (with at least 50% taxon occupancy). Note that the numbers depicted in panel A reflect the sum of the numbers of SC-OGs and MC-OGs in panels B and C. The data underlying this figure can be found in figshare (doi: 10.6084/m9.figshare.16875904).
(TIF)

**S2 Fig. The number of SNAP-OGs identified in orthologous groups of genes with 2 or more homologs in 1 or more species.** The number of SNAP-OGs per orthologous group of genes is depicted on the x-axis. For example, in the budding yeasts dataset, 977 gene families had 1 SNAP-OG each. The highest number of SNAP-OGs identified in a single orthologous group of genes in each dataset were as follows: in budding yeasts, 5 SNAP-OGs were identified in 1 orthologous group of genes that encode transcriptional activators; in filamentous fungi, 5 SNAP-OGs were identified in each of 2 orthologous groups of genes that encode multifacilitator superfamily transporters and amino acid permeases; and in mammals, 4 SNAP-OGs were identified in each of 3 orthologous groups of genes that encode voltage-gated potassium channels, casein kinases, and a tropomyosin family of actin-binding proteins. The data underlying this figure can be found in figshare (doi: 10.6084/m9.figshare.16875904).
(TIF)

**S3 Fig. The 10 most frequent best-fitting substitutions models are similar between SC-OGs and SNAP-OGs.** The top 10 most frequently observed best-fitting substitutions models were similar between SC-OGs and SNAP-OGs among (**A**) 1,668 SC-OGs and 1,392 SNAP-OGs in budding yeasts, (**B**) 4,393 SC-OGs and 2,035 SNAP-OGs in filamentous fungi, and (**C**) 321 SC-OGs and 1,775 SNAP-OGs in mammals. For example, the LG+F+I+G4 model was the most frequently observed best-fitting substitution model in SC-OGs and SNAP-OGs from budding yeasts. The data underlying this figure can be found in figshare (doi: 10.6084/m9.figshare.16875904).
(TIF)

**S4 Fig. Distributions of information content among SNAP-OGs and SC-OGs.** Boxplot and violin plot distributions of 9 properties representative of phylogenetic information are depicted SNAP-OGs (blue) and SC-OGs (orange) in the (**A**) 1,668 SC-OGs and 1,392 SNAP-OGs in budding yeasts, (**B**) 4,393 SC-OGs and 2,035 SNAP-OGs in filamentous fungi, and (**C**) 321 SC-OGs and 1,775 SNAP-OGs in mammals. Abbreviations are as follows: average bootstrap support (ABS), degree of violation of the molecular clock (DVMC), relative composition variability, Robinson-Foulds distance (RF distance), alignment length (Aln. len.), the number of parsimony informative sites (PI sites), saturation, treeness (tness), and treeness/RCV (tness/RCV). The data underlying this figure can be found in figshare (doi: 10.6084/m9.figshare.16875904).
(TIF)

**S5 Fig. Quality of representation and contributions of properties of phylogenetic information content during principal component analysis.** Principal component analysis was used

to qualitatively compare the similarities and differences between SNAP-OGs and SC-OGs (Fig 3). The leftmost figure in each panel of budding yeasts (**A**), filamentous fungi (**B**), and mammals (**C**) represents the quality of representation for each property across all principal components. The next 2 figures depict the contribution of each property (or variable) to the first and second dimension in reduced dimensional space. The red dashed line represents equal contributions from each variable. The data underlying this figure can be found in figshare (doi: 10. 6084/m9.figshare.16875904).
(TIF)

**S6 Fig. The number of SNAP-OGs identified in an orthologous group of genes with 2 or more homologs in 1 or more species for the dataset used to examine a contentious branch in the tree of life.** The number of SNAP-OGs per orthologous group of genes is depicted on the x-axis. For example, a single SNAP-OG was identified in 1,330 gene families with 2 or more homologs in 1 or more species, whereas 4 SNAP-OGs were identified in 2 gene families with 2 or more homologs in 1 or more species. The data underlying this figure can be found in figshare (doi: 10.6084/m9.figshare.16875904).
(TIF)

**S7 Fig. The 10 most frequently observed best-fitting substitutions models are similar between SC-OGs and SNAP-OGs in the dataset used to examine a contentious branch in the tree of life.** Similar best-fitting substitutions models were observed between 252 SC-OGs and 1,428 SNAP-OGs in a dataset of mammals, which was used to investigate patterns of support in a contentious branch in the tree of life concerning deep evolutionary relationships among placental mammals. The data underlying this figure can be found in figshare (doi: 10. 6084/m9.figshare.16875904).
(TIF)

**S8 Fig. Cartoon comparison of different tree decomposition algorithms.** Using the phylogeny presented in Fig 1B (panel A) and Fig 2B (panel B), different tree decomposition algorithms are compared. (**A**) OrthoSNAP will identify 4 SNAP-OGs, whereas DISCO and the maximally inclusive strategies will each identify 3 subgroups of orthologous genes. PhyloTree-Pruner will not identify any subgroups of single-copy orthologous genes. (**B**) OrthoSNAP will identify 5 subgroups of single-copy orthologous genes (light blue) by identifying maximally inclusive subgroups—subtrees where each taxon is represented by a single sequence—and maximally inclusive subgroups after species-specific inparalog trimming (species-specific inparalogs are shown in orange). In contrast, DISCO and maximally inclusive strategies will identify 3 SC-OGs, in part, because they do not account for species-specific inparalogs. PhyloTreePruner, which only prunes species-specific inparalogs, will not identify any subgroups of single-copy orthologous genes due to the presence of more ancient duplication events.
(TIF)

**S1 Table. Species and accession numbers for proteomes used in each dataset.** This table details the species used for the budding yeasts, filamentous fungi, and mammalian datasets. All proteomes from budding yeasts were downloaded from Shen and colleagues [51]. Proteomes from filamentous fungi and mammals were downloaded from NCBI, and their accessions and assembly names are provided.
(XLSX)

**S2 Table. Number of orthogroups examined.** A table of the number of orthogroups, the number of SC-OGs, the number of gene families with orthologs and paralogs (MC-OGs), and

the number of SNAP-OGs examined in the present study.
(XLSX)

**S3 Table. Ortholog occupancy for each dataset.** A table summarizing the average and standard deviation of taxon completeness in SC-OGs and SNAP-OGs.
(XLSX)

**S4 Table. Nine properties of phylogenetic information content.** Phylogenetic information content of SC-OGs and SNAP-OGs were examined using the 9 properties described here. The abbreviation, description, additional notes, and function in PhyKIT used to calculate each property are listed here.
(XLSX)

**S5 Table. Multifactor analysis of variance results reveals no substantial differences between SC-OGs and SNAP-OGs.** Degree of freedom, sum of squares, mean square, F-value, and *p*-value for multifactorial analysis of variance are shown here. Multifactorial analysis of variance was conducting accounting for potential interaction effects as well as using an additive model, which does not account for interaction effects.
(XLSX)

**S6 Table. Tree certainty and tree certainty-all results.** Examining tree certainty and tree certainty-all revealed similar levels of incongruence among gene trees inferred using SC-OGs and SNAP-OGs.
(XLSX)

**S7 Table. Dataset for examining deep evolutionary relationships among eutherian mammals.** The NCBI accession, assembly name, name in files, and ingroup/outgroup designations are detailed here for each proteome used.
(XLSX)

**S8 Table. Number of orthogroups examined among eutherian mammals.** A table of the number of orthogroups, the number of SC-OGs, the number of gene families with orthologs and paralogs (MC-OGs), and the number of SNAP-OGs examined among eutherian mammals.
(XLSX)

**S9 Table. Gene support frequency results among ancient eutherian mammalian relationships.** Gene support frequency results reveal similar levels of support between the 3 hypotheses concerning deep evolutionary divergences among mammals. Multitest corrected *p*-values are also shown here.
(XLSX)

**S10 Table. Comparison between different algorithms that identify subgroups of orthologous genes or conduct species-specific inparalog trimming.** Notably, OrthoSNAP provides the most user flexibility and handles the most use cases.
(XLSX)

## Acknowledgments

We thank the Rokas lab for helpful discussion and feedback.

## Author Contributions

**Conceptualization:** Jacob L. Steenwyk, Antonis Rokas.

**Data curation:** Jacob L. Steenwyk.

**Formal analysis:** Jacob L. Steenwyk.

**Funding acquisition:** Jacob L. Steenwyk, Antonis Rokas.

**Investigation:** Jacob L. Steenwyk, Dayna C. Goltz, Thomas J. Buida, III, Yuanning Li, Xing-Xing Shen.

**Methodology:** Jacob L. Steenwyk, Dayna C. Goltz, Thomas J. Buida, III, Yuanning Li, Xing-Xing Shen.

**Project administration:** Antonis Rokas.

**Resources:** Jacob L. Steenwyk, Antonis Rokas.

**Software:** Jacob L. Steenwyk.

**Supervision:** Antonis Rokas.

**Validation:** Jacob L. Steenwyk.

**Visualization:** Jacob L. Steenwyk.

**Writing – original draft:** Jacob L. Steenwyk, Antonis Rokas.

**Writing – review & editing:** Jacob L. Steenwyk, Dayna C. Goltz, Thomas J. Buida, III, Yuanning Li, Xing-Xing Shen.

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
