## [Editor Report · Decision Letter 0]

9 Nov 2021

Dear Antonis, 

Thank you for submitting your manuscript entitled "­­orthoSNAP: a tree splitting and pruning algorithm for retrieving single-copy orthologs from gene family trees" for consideration as a Methods and Resources paper by PLOS Biology.

Your manuscript has now been evaluated by the PLOS Biology editorial staff, as well as by an academic editor with relevant expertise, and I'm writing to let you know that we would like to send your submission out for external peer review.

Once your full submission is complete, your paper will undergo a series of checks in preparation for peer review. Once your manuscript has passed the checks it will be sent out for review. 

If your manuscript has been previously reviewed at another journal, PLOS Biology is willing to work with those reviews in order to avoid re-starting the process. Submission of the previous reviews is entirely optional and our ability to use them effectively will depend on the willingness of the previous journal to confirm the content of the reports and share the reviewer identities. Please note that we reserve the right to invite additional reviewers if we consider that additional/independent reviewers are needed, although we aim to avoid this as far as possible. In our experience, working with previous reviews does save time. 

If you would like to send your previous reviewer reports to us, please specify this in the cover letter, mentioning the name of the previous journal and the manuscript ID the study was given, and include a point-by-point response to reviewers that details how you have or plan to address the reviewers' concerns. Please contact me at the email that can be found below my signature if you have questions. 

Please re-submit your manuscript within two working days, i.e. by Nov 11 2021 11:59PM.

Kind regards,

Roli

Roland Roberts

Senior Editor

PLOS Biology

rroberts@plos.org

---

## [Decision Letter · Decision Letter 1]

17 Jan 2022

Dear Dr Rokas,

Thank you for submitting your manuscript "­­orthoSNAP: a tree splitting and pruning algorithm for retrieving single-copy orthologs from gene family trees" for consideration as a Methods and Resources at PLOS Biology. Your manuscript has been evaluated by the PLOS Biology editors, an Academic Editor with relevant expertise, and by three independent reviewers.

You'll see that each of the reviewers is broadly positive about your study, but each raises a number of concerns that need to be addressed before further consideration. For example, you’ll see that Reviewer #1 wants you to clarify how the method differs from others, Reviewer #2 worries that your method is simplistic and might break down when faced with more complex scenarios (and wants to be convinced otherwise), and Reviewer #3 found orthoSNAP easy to use, but queries the decision to always retain the longest orthologue, questions the power of your stats, and wonders whether you could incorporate synteny.

In light of the reviews (below), we will not be able to accept the current version of the manuscript, but we would welcome re-submission of a much-revised version that takes into account the reviewers' comments. We cannot make any decision about publication until we have seen the revised manuscript and your response to the reviewers' comments. Your revised manuscript is also likely to be sent for further evaluation by the reviewers.

We expect to receive your revised manuscript within 3 months. 

**IMPORTANT - SUBMITTING YOUR REVISION**

*Re-submission Checklist*

*Published Peer Review*

*PLOS Data Policy*

*Blot and Gel Data Policy*

Sincerely,

Roli Roberts

Roland Roberts

Senior Editor

PLOS Biology

rroberts@plos.org

REVIEWERS' COMMENTS:

Reviewer #1:

[identifies himself as Yannis Nevers]

The authors present OrthoSNAP, a method that automatically extracts marker genes (a set of orthologous sequences with representatives from most species in a given dataset) from a set of gene families and their corresponding gene trees.

Contrary to more traditional method; it does not focus on single copy orthologous groups (sc-OGs) and also extracts orthologous groups from gene families even in presence of duplication (termed SNAP-OGs). Using SNAP-OGs, and not only Sc-OG, allows to increase the number of gene markers one can extract from a set species, to later use in phylogenomic analyses.

The authors show, comparing it over several descriptors, that the phylogenetic signal in SNAP-OGs is not significantly different than one in Sc-OG. They also show in a specific case that using SNAP-OGs for species tree determination on a difficult to infer taxonomic node yield similar results than using sc-OG, suggesting SNAP-OGs could be used the same way sc-OGs are. The analyses are convincing and to my knowledge, one of the first confirmation that SNAP-OGs can be used on the same capacity of Sc-OGs in phylogenomic analysis.

The manuscript is well organized and clearly written. My review below is divided into sections relative to different aspects of the work, and concludes with my recommandations.

NOVELTY:

SNAP-OG is based on a tree splitting and pruning strategy, used to isolate OGs nested into gene families that underwent ancient duplications and keep only one copies from recently duplicated species-specific inparalog. Generally, the aim is to extract a set of orthologous sequences with only one representative of most of the species of interest, even in presence of duplication events.

As the authors discuss in the introduction, this is not per se new in concept. In particular, methods like PhyloTreepruner and Agalma have similar aims and can at least be used to handle species-specific inparalogs.

The authors discuss this and claim their method handles more types of paralogy than previously existing methods, and to my understanding, can extract more OGs from a set of gene families with duplication events than these methods could (presumably because of the combination of tree splitting and pruning that is implemented in OrthoSNAP). Still, it is unclear from the current discussion exactly how they differ, and in what concrete case OrthoSNAP yields more orthologous groups than these methods.

Additionally, two other published works appear to implement similar strategies to extract OGs from gene families with duplication: DISCO (Willson et al, 2021) [doi: 10.1093/sysbio/syab070] and the different strategies of orthologs sampling proposed in Yang and Smith, 20145] [doi:10.1093/molbev/msu24 (in particular the Maximum inclusion strategy). These works are cited in the manuscript but there is no discussion relative to the strategy they use for what looks like to be a similar task.

Finally, I note that SNAP-OGs provided by OrthoSNAP seems to be conceptually similar to OMA Groups provided by the OMA Orthology inference method (see Identifying orthologs with OMA: A primer, Wahn-Zabal et al, doi: 10.12688/f1000research.21508.1). Do the authors believe it is the case? If so, it may be worth mentioning.

While I believe these aspects may be discussed a bit more (see recommendations), OrthoSNAP does seem to implement a different strategy than existing methods and the fact that it is implemented as an user-friendly command line tool with few input constraints is valuable.

The assessment of the difference between sc-OGs and SNAP-OGs is both new and has interesting implications regarding the use of SNAP-OGs in phylogenomic studies. It has a similar conclusion than another independent but somewhat similar preprinted studies. This is also addressed by the authors in the discussion section.

SOFTWARE ACCESSIBILITY, EASE OF USE AND REPRODUCIBILITY

The code is available on GitHub, and is well documented. The documentation provided an easy and quick way to install the software and was sufficient for me to successfully run it on a test dataset. 

The datasets used in the course of the work are freely available on figshare and the protocole is sufficiently described in the method section

RECOMMENDATION:

I believe the work would be of interest for the community, in good part because it provides easy to use software for the extraction of marker genes from phylogenomic datasets, even in presence of duplication. I thus recommend its publication without additional analyses needed.

However, given the similarity of the work with previously available software, I'd like to see a more detailed discussion on how OrthoSNAP differs from previous works. In particular:

-The author indicates that OrthoSNAP is different from two other tree pruning protocole. It would be useful to indicate a few example cases of when it is expected to provide different results.

-I would also be interested in a discussion of the comparison with other published strategies for what seems to be a similar task: in particular the Maximum inclusion strategy from Yang et al, 2014, the methods used by DISCO. (See the Novelty section above)

Minor remark:

-The authors use the word "taxon" to refer only to the individual species in their datasets. It causes a bit of confusion because usually any grouping of multiple species sharing a common ancestor may be considered a taxon. I suggest the authors disambiguate the term.

Reviewer #2:

The authors present here orthoSNAP a program designed to obtain orthogroups from gene trees including duplications with the objective of obtaining more marker genes and therefore increasing the number of genes available for numerous evolutionary studies. As stated by the authors, this has been done before in multiple ways and the improvement given by orthoSNAP is more focused on usability by non-expert users than on creating an innovative algorithm. They then test their orthoSNAP on different sets of species: budding yeasts, filamentous fungi and a set of mammals. They show how using orthoSNAP they can increase the number of orthogroups significantly depending on the dataset and that those orthogroups are not different in terms of general phylogenetic characteristics making them suitable for posterior analyses. 

I have concerns that the orthoSNAP algorithm is a bit too simple and will not hold up in the case of more complex scenarios (note also my comments on the datasets below). A very obvious scenario in which I think orthoSNAP will fail is when duplications are concentrated in a given node. For instance, imagine that the budding yeast dataset was formed of more than 50% species that underwent the WGD that happened in the Saccharomyces cerevisiae lineage and the rest was made of pre-WGD species. As I understand, the algorithm would look for clades that include more than 50% of the species set and no duplications after removing species specific duplicates. In the dataset I proposed you would have numerous orthogroups that would include uniquely post-WGD species and would not include the pre-WGD orthologs despite existing, being present in the tree and being easily detectable. I wonder if the taxon distribution between SC-OGs and SNAP-OGs would still be the same and how the other metrics would be affected. The same problem can be extended to any tree analysed, orthologs found before a duplication large enough to cope the taxon occupancy threshold will not be included in orthogroups. 

Note that in the introduction the authors used plants as an example of a complicated dataset where tools such as orthoSNAP would be of great advantage, yet the authors used two fungal datasets for which they already have a significant amount of single copy families. This is great for showing whether there are differences between SC-OGs and SNAP-OGs, but they do not offer a true challenge for the program. The mammal dataset is more meaningful in terms of what orthoSNAP can offer and I think there should be more emphasis put on this.

Regarding the mammal dataset, the improvement is notable and it certainly is useful but I miss a bit more of context on why of the 17407 multi copy families orthoSNAP only retrieves 1775. 

Regarding the comparison between SC-OGs and SNAP-OGs, I understand that the underlying point of the analysis is to show that orthogroups build from single copy genes and from SNAP will give similar results and that it is so because orthoSNAP is able to successfully capture only orthologs and not include unwanted paralogs. For that point to be meaningful, you should also show that randomly including paralogs would be detrimental to the metrics you are using and would therefore give a different signal in these specific datasets. If not added, one could argue that, no matter how well or poorly SNAP performs the results are always going to be the same. Points also to see whether mixing different kinds of paralogs have different effects.

Minor comments:

1.- When having a species specific duplication orthoSNAP keeps only the gene with the longest sequence, argument based on transcriptomic data. I would be interested in reading why authors did not choose to keep the paralog with the shortest branch as that would be more conservative when thinking on running some of the posterior analyses.

2.- In the results section I miss having plain numbers shown about the real improvement of orthoSNAP. It is put as a percentage and directed to a supplementary table but I think the number of orthogroups retrieved for each dataset should be put there.

3.- It would also be interesting to have more of an idea from the start which species are used in the analysis. Filamentous fungi is not the same as saying a group of Aspergillus and Penicillium genomes that are very closely related.

4.- I wonder how orthoSNAP would perform without having the pre-filter of orthofinder. Orthofinder is already putting genes into orthologous families but one could build a tree based on blast results and just run orthoSNAP on that. Would results be comparable?

Reviewer #3:

[identify themselves as Giulio Formenti and Erich Jarvis]

General comments:

Steenwyk et al. present orthoSNAP, a tree splitting and pruning algorithm for retrieving single-copy orthologs from gene family trees. The tool is aimed at addressing the important issue in comparative genomics of identifying unique orthologous genes for evolutionary inference.

The logic behind the tool is clear, well-explained, and well-detailed. The tool is well-documented, and we were easily able to install it (on Mac) and run it using the test data set and the tutorial provided. The authors also use orthoSNAP to explore two contentious alternative hypotheses of deep evolutionary relationships among placental mammals. The good news is that from 46,645 orthologous groups of genes, their method identified 6400 more single copy orthologs in addition to the 6600 by another method. My worrisome news is what the many other putative ortholos of the ~46,000?. Perhaps we am missing something here? 

Besides this, we have three main concerns, one regarding the simplicity of a key step of the pipeline, another regarding statistics, and a third regarding complementary evidence, such as synteny.

With respect to simplicity: 

At the pruning step, the paralog to be retained is always the longest. The justification given for this choice is that it is common practice in transcriptomics. I can see why this is the case in transcriptomics, e.g. to increase mappability to the genome, however in the alignment of ortholog genes, choosing the longest transcript this may not be the best choice and could rather introduce bias in the results to a particular spliced exon missing in some of the longest transcript or additional erroneously annotated codons. I wonder if this very simple assumption could undermine some of the results. Can complementary/alternative method using all transcripts, be envisaged and introduced?

With respect to statistics:

Throughout the manuscript, the absence of statistical significance is generally given as evidence to conclude that identified SNAP-OGs behave similarly to SG-OGs. Since the absence of significance can be due to no real significance or not enough power, these results and their interpretation should be phrased more carefully. Additionally, to some degree, one would expect genes with high numbers of paralogs to behave somewhat differently from truly single-copy orthologs; there seems to be some evidence that this is happening, which would be interesting to characterize since, as the authors put it, "the phylogenetic information content of these gene families remains unknown". In particular:

- It seems that the non-significant results in the multivariate analyses may be partially explained by the high dispersion of the data, including in at least some cases the presence of many outliers (e.g. in alignment length, parsimony informative sites, RCV and treeness show higher SD in mammalian SNAP-OGs Fig S9). Are there at least some genes that behave differently? Can the authors comment on that? For instance, in Budding yeast it seems that tree certainty is consistently lower in SNAP-OGs vs SG-OGs (ST6). Can the authors further describe ST6 and comment on this in the main text?

- In Fig 4B, can the authors explain why the added a third hypothesis when testing the evolutionary relationships in placental mammals ("Xenarthra as sister to all other Eutheria represented in yellow")? There is a discrepancy in the main text, as initially only two hypotheses are mentioned, but then a third is introduced and used for only one of the analyses. A third hypothesis will obviously reduce the statistical significance of the results after correction for multiple testing, potentially making results non-significant, which seems to be the case here. Also, why was the alpha set at 0.01 instead of the usual 0.05? Finally, do you think that more sophisticated statistical tests than Fisher's using GSF values that would be able to reveal differences?

- In Fig 4C, the outliers collapse most of the central points. However, the two distributions do look different. Aside from the absence of significant differences in GLS, which could be the result of the great dispersion of the data, could a test that compares the shape of the distribution reveal a significant difference? Also, Fig 4C should be bigger to better highlight the distribution? Maybe a transformation could also be applied to the y axis to better highlight the data.

With respect to synteny:

Because of different rates of difference, sometimes it is difficult to indentify orthologs versus paralogs based on sequence identity and phylogeny alone. Synteny is a third piece evidence that helps identify orthology. A good recent example on the oxytocin receptor and ligand family of genes (Theofanopoulou et al 2021 Nature). Have the authors utilized synteny? If not, it would be good to incorporate. At a minimum the authors should discuss this issue.

Specific comments:

Abstract:

- consider dropping "selection"", as negative selection also needs SC_OGs

- consider dropping "in contrast", since there is no contrast with the previous sentence

- consider spelling "SNAP" in the software name, also to provide clues on its mode of operation, rather than providing a definition of SNAP-OGs directly in the abstract.

Figure 1. Given that one of the claims the authors make in the discussion is that orthoSNAP, as opposed to other tools, can distinguish multiple paralog classes, a more complete explanation of such classes would greatly help the reader. It would then be probably more useful to provide a cartoon tailored to the classification made by orthoSNAP, rather than a general framework. This figure could then be part of the results; or combined with the current figure 1.

Last sentence of first paragraph of the introduction: The availability of more genomes actually make phylogenetic inference and thus ortholgy of genes easier rather than more complex or difficult. Could this sentence be rephrased to account for this?

End of second paragraph of introduction: Rather than broad, I'd say that the past algorithms were not designed to retrieve homologs in large gene families.

The final paragraph of the introduction should be part of the results, and a brief 1-3 sentences stated about the results should be what is in the introduction.

Results, first paragraph. Do the authors have a suggested reason as to why there is variation in the results depending on lineage? It is technical or biological reason?

"Similar to taxon occupancy…." since a 50% cutoff was applied by design to both SC-OGs and SNAP-OGs, isn't this expected?

Second to last sentence of same paragraph: In terms congruence, in contradiction to this, in Budding yeast it seems that tree certainty is consistently lower in SNAP-OGs vs SG-OGs (ST6).

"Resolution of this conflict": It does not look like this analysis is aimed at resolving said conflict on phylogeny, but rather to demonstrate no difference between SNAP-OGs and SC-OGs results. Please rephrase

"More specifically, no differences…" this value is close to a 0.05 significance, suggesting that collapsing branches with low support values starts to highlight differences between SC-OGs and SNAP-OGs. I think these differences could be valuable to understand the nature of SNAP-OGs and deserve further analysis. Why use Benjamini-Hochberg correction?

Last paragraph of results: Why is this striking? One could expect that in difficult-to-resolve branches the noise would be even higher, reducing our power to detect significant differences

Discussion. First 1.5 paragraphs are repetitive of the introduction. They can be deleted.

In certain datasets, .. This only refers to mammals. I would therefore specify

Can the authors suggest a possible interpretation of why SNAP-ORG were five times more prevalent than SC-OGs?

Qualitatively similar: Can the authors clarify what does qualitatively similar mean? 

We also note that our algorithm: This is interesting, however it is the first time outparalogs, inparalogs, etc are mentioned (besides figure 1 legend). It does not appear to be mentioned in Methods either. More details would be useful.

Three noteworthy differences: these differences are not in the results but in the approach, please clarify.

Methods, Dataset generation: This is the first time that it is made clear that the analysis is conducted on protein sequences rather than on DNA sequences (which is implicit when talking about genes). This is an important point and should be made immediately clear, first time in the abstract. 

Supplementary figures:

- Fig S1 Information content is minimal and fully overlaps with the text, therefore it doesn't seem a necessary figure.

- Fig S2 can scales be reported directly in axis ticks rather than in axis titles.

- Fig S7 Similar to Fig S1, the content is minimal and fully overlaps with the main text, therefore it doesn't seem a necessary figure.

---

## [Decision Letter · Decision Letter 2]

9 Sep 2022

Dear Dr Rokas,

Thank you for your patience while we considered your revised manuscript "­­OrthoSNAP: a tree splitting and pruning algorithm for retrieving single-copy orthologs from gene family trees" for publication as a Methods and Resources at PLOS Biology. This revised version of your manuscript has been evaluated by the PLOS Biology editors, the Academic Editor, and two of the original reviewers.

Based on the reviews, we are likely to accept this manuscript for publication, provided you satisfactorily address the remaining points raised by the reviewers. Please also make sure to address the following data and other policy-related requests.

IMPORTANT: Please attend to the following:

a) Please address the remaining minor requests from the two reviewers.

b) Please address my Data Policy requests below; specifically, we need you to supply the numerical values underlying Figs 3ABC, 4BC, S1ABCD, S2, S3ABC, S4ABC, S5ABC, S6, S7, S8B, either as a supplementary data file or as a permanent DOI’d deposition, e.g. part of your Figshare depo (my understanding from the Figshare DOI is that this contains the raw input data, rather than the data presented in these Figure panels; please clarify).

c) Please cite the location of the data clearly in all relevant main and supplementary Figure legends, e.g. “The data underlying this Figure can be found in S1 Data” or “The data underlying this Figure can be found in https://figshareXXXXX”

We expect to receive your revised manuscript within two weeks. 

*Published Peer Review History*

*Press*

Sincerely,

Roli Roberts

Roland Roberts, PhD

Senior Editor,

rroberts@plos.org,

PLOS Biology

DATA POLICY:

Regardless of the method selected, please ensure that you provide the individual numerical values that underlie the summary data displayed in the following figure panels as they are essential for readers to assess your analysis and to reproduce it: Figs 3ABC, 4BC, S1ABCD, S2, S3ABC, S4ABC, S5ABC, S6, S7, S8B. NOTE: the numerical data provided should include all replicates AND the way in which the plotted mean and errors were derived (it should not present only the mean/average values).

DATA NOT SHOWN?

REVIEWERS' COMMENTS:

Reviewer #2:

The authors have submitted a very thorough revision of their manuscript. I think it now covers what was required from my side. Just two minor comments:

In the sentence "30 plants, which are known to complex histories of gene duplication and loss" should be "known to have"

They mention: "In comparison, 15 SC-OGs were identified in the plant dataset; 2,782 in the budding yeast dataset; and 390 in the choanoflagellate dataset", there are more SC-OGs in the yeast dataset than SNAP-OGs, could that be a typo? If not, what is the explanation for this?

Reviewer #3:

[identifies himself as Erich Jarvis]

The authors were very responsive to the reviews, and have made substantial improvements to the manuscript. This includes performing many additional analyses. All of our main concerns have been satisfied. Just two clarifications needed. 

The authors stated that OMA is sequence based, whereas OrthoSNAP is phylogeny based. But isn't the phylogeny of the genes based on sequence alignments?

The explanation of lineage sorting for not identifying all one-to-one orthologs makes sense. The authors can cite some papers where they show a high proportion of genes (5-30%) that can have incomplete lineage sorting between species (e.g. Jarvis et al 2014 Science).

---

## [Editor Report · Decision Letter 3]

13 Sep 2022

Dear Dr Rokas,

Thank you for the submission of your revised Methods and Resources "­­OrthoSNAP: a tree splitting and pruning algorithm for retrieving single-copy orthologs from gene family trees" for publication in PLOS Biology. On behalf of my colleagues and the Academic Editor, Andreas Hejnol, I'm pleased to say that we can in principle accept your manuscript for publication, provided you address any remaining formatting and reporting issues. These will be detailed in an email you should receive within 2-3 business days from our colleagues in the journal operations team; no action is required from you until then. Please note that we will not be able to formally accept your manuscript and schedule it for publication until you have completed any requested changes.

Sincerely, 

Roli Roberts

Senior Editor

PLOS Biology

rroberts@plos.org